5

# Long-term meteorological and carbon, water and energy flux data from the Boreal Ecosystem Research and Monitoring Sites, Saskatchewan, Canada

Alan Barr<sup>1,2</sup>, T. Andrew Black<sup>3</sup>, Warren Helgason<sup>4,1,2</sup>, Andrew Ireson<sup>5,1</sup>, Bruce Johnson<sup>1</sup>, J. Harry McCaughey<sup>6,†</sup>, Zoran Nesic<sup>3</sup>, Charmaine Hrynkiw<sup>7</sup>, Amber Ross<sup>7</sup>, Newell Hedstrom<sup>7</sup>

<sup>1</sup>Global Institute for Water Security, University of Saskatchewan, Saskatoon, S7N 3H5, Canada
 <sup>2</sup>Centre for Hydrology, University of Saskatchewan, Saskatoon, S7N 5C8, Canada
 <sup>3</sup>Faculty of Land and Food Systems, University of British Columbia, Vancouver, V6T 1Z4, Canada

<sup>4</sup> Civil, Geological, and Environmental Engineering, University of Saskatchewan, Saskatoon, S7N 5A9, Canada
 <sup>5</sup>School of Environment and Sustainability, University of Saskatchewan, Saskatoon, S7N 5C8, Canada
 <sup>6</sup> Department of Geography and Planning, Queen's University, Kingston, K7L 3N6, Canada
 <sup>7</sup> Environment and Climate Change Canada, Saskatoon, S7N3H5, Canada
 <sup>†</sup>deceased

Correspondence: Warren Helgason (warren.helgason@usask.ca)

**Abstract.** The Boreal Ecosystem Research and Monitoring Sites (BERMS) are a network of flux tower research sites located near the southern boundary of the Boreal Plains Ecozone in Saskatchewan, Canada. This network includes four principal sites that characterize the region's dominant vegetation types: mature trembling aspen (Old Aspen, OA, 1997-2017), mature black spruce (Old Black Spruce, OBS, 1997-present), mature jack pine (Old Jack Pine, OJP,

- 1997-present), and a minerotrophic patterned fen (Fen, 2002-present). The dataset reported here include continuous long-term records of site meteorological variables (air temperature, humidity, barometric pressure, precipitation, wind speed and direction), vertical profiles of soil temperature and volumetric water content, surface energy balance components (soil and biomass heat fluxes, photosynthetic heat flux, and eddy covariance-derived latent and sensible heat fluxes), and carbon fluxes (net ecosystem production, gross primary productivity, and ecosystem respiration).
- The strengths of the data set are its length and completeness, spanning up to 27 years; the care given to the measurement of net radiation and the minor surface energy balance terms; the care given to the measurement of precipitation and other hydrologic variables; and the proximity of the sites, which enables inter-site comparisons of the responses of the carbon and water balances to climatic controls. The data are available at <a href="https://doi.org/10.20383/103.01318">https://doi.org/10.20383/103.01318</a> (Helgason et al., 2024).

# 30 1 Introduction

The Boreal Ecosystem Research and Monitoring Sites (BERMS) are a network of flux-tower research sites located near the southern edge of the Boreal Plains Ecozone in Saskatchewan, Canada. The BERMS sites were established in 1993 as part of the Southern Study Area of the Boreal Ecosystem-Atmosphere Study (BOREAS) (Sellers et al., 1997). The primary BOREAS objectives were to better understand of the role of the boreal forest in the global carbon

cycle and climate system; to create integrated data sets that could be used to evaluate and improve climate, hydrologic,

and ecosystem C-cycle models; and to better anticipate the potential effects of climate change on this important biome. The BOREAS data from 1994-1996 are available at the Oak Ridge National Laboratory Distributed Active Archive Center (ORNL DAAC). The BERMS program has benefitted greatly from BOREAS' multidisciplinary design, its integration of measurements with modelling, and the priority given to data management. When BOREAS ended in

- 1996, the Southern Old Aspen (OA), Old Black Spruce (OBS), and Old Jack Pine (OJP) flux towers continued as BERMS, and the Southern Fen flux tower was restarted in 2002. BERMS played a leading role in the Fluxnet-Canada Research Network (FCRN, 2002-2007) and the Canadian Carbon Program (2007-2011), and has contributed to the Changing Cold Regions Network (2012-2017), Global Water Futures (2017-2023), the Global Water Futures Observatories (starting 2023), the North American Carbon Program, AmeriFlux, and FLUXNET.
- The BERMS program was originally operated and funded by Environment and Climate Change Canada, Natural Resources Canada, and Parks Canada, in partnership with researchers at the University of British Columbia (UBC), Queen's University, and the University of Saskatchewan. Since 2012, the Global Institute for Water Security (GIWS), University of Saskatchewan, has led the program, with strong continuing involvement of UBC at OA and OBS through 2017. After 2017, the OA site was decommissioned, and GIWS assumed full responsibility for operations at OBS, 50 OJP, and Fen.
  - The Boreal Plains Ecozone covers 701,750 km<sup>2</sup> (7% of Canada's land area). It extends from northeastern British Columbia, across portions of Alberta and central Saskatchewan, to Lake Winnipeg in Manitoba. The ecozone is characterized by a northern continental climate, with long, cold winters and short cool summers. Over 60% of the Boreal Plains is forested including coniferous evergreen needleleaf (42%), broadleaf deciduous (37%), and mixed
- forests (20%), with the topography typically characterized by a flat to gently rolling, hummocky and kettled terrain (ESTR Secretariat, 2014). The Boreal Plains is a diverse land-cover mosaic with a spatial pattern of climax vegetation controlled by soil drainage and available soil moisture (Ireson et al., 2015). In general, jack pine forests are located on the well-drained sandy uplands, mixed-wood and aspen forests on the well-drained loamy uplands, black spruce forests and treed wetlands in the lowland depressions, and peatlands in the poorly-drained deeper depressions. The
- four long-term BERMS sites represent the dominant vegetation types in the Boreal Plains Ecozone, including mature forest stands of trembling aspen (Old Aspen OA), black spruce (Old Black Spruce OBS), and jack pine (Old Jack Pine OJP), and a minerotrophic patterned fen (Fen).

The BERMS sites are strategically located near the southern edge of the western boreal forest, a region that is expected to be highly sensitive to climate change (Ireson et al., 2015). The geographic transition from grasslands and croplands

(to the south) to boreal forest (to the north) coincides with a shift in the water balance, with diminishing water deficits to the north (Hogg, 1997). If future climate warming in this region is associated with more frequent and severe water deficits, as is likely to be the case, the southern edge of the boreal forest is expected to shift northward, with biome shifts occurring primarily after wildfire (Ireson et al., 2015).

The BERMS data sets have been widely used, in better understanding the ecophysiological processes that control the exchanges of carbon, water and energy between the ecosystem and the atmosphere (e.g., Liu et al., 2019; Liu et al., 2022), in synthesis studies (e.g. Ireson et al., 2015; Helbig et al., 2020), for model evaluation and improvement (e.g. Keenan et al., 2012; Richardson et al., 2012), and as ground-truth in the development of remote sensing methods (e.g.

Lambert et al., 2013; Pulliainen et al., 2017). The strengths of the data include: its length and completeness, spanning over 25 years; the consistency in instrumentation and data processing; the care given to the measurement of net

radiation and the minor energy balance terms; the care given to the measurement of precipitation and other hydrologic variables, including streamflow from the White Gull Creek watershed containing the OBS and OJP sites; and the close proximity of the sites to each other, which enables intercomparison of the responses of different vegetation types to climatic forcings.

This paper documents the core flux and meteorological data sets at the four long-term BERMS sites. The objectives

are to provide background information about the sites; to document the data sets including instrumentation and datalogging history, data post-processing, quality assurance, and gap-filling methods; to assess data quality; to identify a few known problems; and describe if and how the problems have been resolved. The data are archived through the Federal Research Data Repository of the Digital Research Alliance of Canada and are available at <u>https://doi.org/10.20383/103.01318</u> (Helgason et al., 2024). Additional measurements not reported here include water

table depth, snow depth, and periodic surveys of snow water equivalent and tree biometry.

# 2 Study Area

The BERMS study area is in the Boreal Plains Ecozone, near the southern edge of the boreal forest (Fig. 1). The continental climate is characterized by having a long and dry cold season and a short growing season. The most recent climate normals for nearby climate stations are for Waskesiu Lake (53.92 °N, -106.07 °W, 1971-2000), with mean

annual, January and July air temperatures of 0.4, -17.9 and 16.2 °C, respectively, and mean annual precipitation of 467 mm, 30% of which fell as snow; and Prince Albert Airport (53.22 °N, -105.67 °W, 1991-2020), with mean annual, January and July air temperatures of 1.4, -17.2 and 17.9 °C, respectively, and mean annual precipitation of 432 mm (1991-2020), 27% of which fell as snow (based on the 1981-2010 normals).

The eastern portion of the BERMS study area includes the White Gull Creek watershed (603 km<sup>2</sup>), which has been gauged for continuous streamflow measurement since 1993 (Water Survey of Canada, 2024), annual streamflow, 1994-2022, corresponding to a mean ( $\pm$  s.d.) runoff depth of 125  $\pm$  64 mm y<sup>-1</sup>.

Figure 1. Location of the Old Aspen (OA), Old Black Spruce (OBS), Old Jack Pine (OJP), and Fen sites in the province of Saskatchewan, Canada.

#### 2.1 Flux-tower sites

The long-term BERMS flux-tower sites include two mature evergreen needleleaf forests (black spruce and jack pine), one deciduous broadleaf forest (trembling aspen) and one wetland fen. Site properties are given in Tables 1 (location and vegetation) and 2 (soils).

#### 2.1.1 Old Aspen

The Old Aspen (OA) site is located in Prince Albert National Park, about 70 km NW of Prince Albert. The forest stand has two distinct layers: an overstory of trembling aspen (*Populus tremuloides* Michx. 95 %) with scattered balsam poplar (*Populus balsamifera* L. 5 %) and a 2-m understory of dense beaked hazelnut (*Corylus cornuta* Marsh.)

- (Griffis et al., 2003, 2004). The stand was established by natural regeneration after a forest fire in 1919 (Kljun et al., 2006), with subsequent fires eliminating the conifer seed source and resulting in a nearly pure deciduous stand (Dave Weir, Prince Albert National Park, *personal communication*). The soil is an Orthic Gray Luvisol (Blanken et al., 1997) having a 10-cm LFH organic horizon overtop a 30-cm silt loam mineral soil horizon overlying clay-rich glacial till subsurface materials. The water table depths range from 1 to 5 m below the ground surface, varying spatially
- according to location in the hummocky terrain, with ponded water in low-lying depressions during wet years. Further description of this site can be found in Blanken et al. (1997).

# 2.1.2 Old Black Spruce

Old Black Spruce (OBS) is located 115 km northeast of Prince Albert, Saskatchewan. The forest stand was established by natural regeneration after a forest fire in 1879 (Kljun et al., 2006, Krishnan et al., 2008). The site is dominated by

- black spruce (*Picea mariana* [Mill.] BSP), but also contains ~10% tamarack (*Larix laricina* Du Roi). The understory consists of sparse shrubs (e.g., *Rhododendron groenlandicum* (Oeder), Kron & Judd and *Vaccinium vitisidaea* L.). The predominant groundcover is feather moss (*Hylocomium splendens* [Hedw.] Schimp., *Pleurozium schreberi* (Brid.) Mitt., with patches of hummocky peat (*Sphagnum* spp.) in wet areas and reindeer lichen (*Cladonia* spp.) in drier areas. The soil is Peaty Orthic Gleysol with a 20-30 cm deep peat layer overlying waterlogged sand, with imperfect to poor
- drainage. The surface is relatively flat with a hummock-hollow micro topography in wetter areas with occasional surface water in the hollows (Jarvis et al., 1997). Depth to water table varies between 0 and 1 m below the ground surface. Further description can be found in Jarvis et al. (1997), Gower et al. (1997), and Gaumont-Guay et al. (2014).

# 2.1.3 Old Jack Pine

Old Jack Pine (OJP) is located on a glacial outwash plain about 140 km northeast of Prince Albert. The dominant tree species is jack pine (*Pinus banksiana* Lamb.). The dominant understory ground cover is reindeer lichen (*Cladonia mitis* [Sandst.] Hustich) with isolated groups of green alder (*Alnus crispa* [Ait.] Pursh) and feathermoss (*Pleurozium* 

spp.) (Gower et al., 1997). The stand was established by natural regeneration after a forest fire in 1914 (Zha at al., 2009). The soil is a degraded Eutric Brunisol/Orthic Eutric Brunisol and the site topography is relatively flat with few gently rolling ridges (Baldocchi et al., 1997). The sandy soil is nutrient poor and well drained with the water table lying at least 5 m below the ground surface. For additional site details, see Baldocchi et al. (1997) and Gower et al.

#### 2.1.4 Fen

(1997).

The Fen site is a moderately rich, minerotrophic patterned fen surrounded by black spruce and jack pine forests, 120 km northeast of Prince Albert, Saskatchewan. The fen is orientated northwest-southeast and contains 8.5 km<sup>2</sup> of

140 peatland (Sonnentag et al., 2010). The peat depth varies from 1 m near the edges up to 2-3 m in the center of the fen. Sedges (*Carex* spp. And *Eriophorum* spp.) and buckbean (*Menyanthes trifoliate* L.) are the dominant herbaceous plants. The dominant woody species include bog birch (*Betula pumila* L.) and tamarack (*Larix laricina* [Du Roi] K. Koch). As of 2019, the bog birch and tamarack species have significantly diminished due to persistent high water levels in the fen. Except for during dry years the water table is generally at or above the peat surface. Further

description of this site is available in Suyker et al. (1997) and Sonnentag et al. (2010).

# Table 1. Characteristics of the BERMS sites

|                       | OA              | OBS          | OJP             | Fen                 |
|-----------------------|-----------------|--------------|-----------------|---------------------|
| Location              | 53.629°N,       | 53.987°N,    | 53.916°N,       | 53.802°N,           |
|                       | 106.198°W       | 105.118°W    | 104.692°W       | 104.618°W           |
| Elevation (m)*        | 596             | 591          | 510             | 485                 |
| Dominant tree species | Trembling aspen | Black spruce | Jack pine       | Scattered bog birch |
|                       |                 |              |                 | and tamarack        |
| Dominant understory   | Beaked hazelnut | Feathermoss  | Reindeer lichen | Buckbean and        |
| species               |                 |              |                 | sedges              |
| Stand height (m)      | 21              | 11           | 14              | n/a                 |
| Stand density (stems  | 964 (1994)      | 5900 (1994)  | 1317 (1994)     | n/a                 |
| ha <sup>-1</sup> )    | 726 (2004)      |              | 1119 (2004)     |                     |
|                       | 542 (2010)      |              | 1061 (2008)     |                     |
|                       | 474 (2016)      |              | 916 (2016)      |                     |
|                       |                 |              | 645 (2023)      |                     |
| Leaf area index**     | 2.4 aspen       | 3.8          | 2.6             |                     |
|                       | 2.0 hazelnut    |              |                 |                     |
| Soil drainage         | Moderate        | Poor         | Good            |                     |

\* The site elevations have been updated based on a recent reassessment and differ from the earlier estimates. In

particular, the earlier value of 579 m at OJP was found to be in error.

\*\* Barr et al. (2004); Chen et al. (2006)

|     | Soil         | Depth          | Bulk                  | Organic C | Soil Texture Class         |
|-----|--------------|----------------|-----------------------|-----------|----------------------------|
|     | Horizon      | (m)            | Density               | Content   | (% Sand, Silt, Clay)       |
|     |              |                | (kg m <sup>-3</sup> ) | (%)       |                            |
| OA  | L            | -0.10 to -0.07 | 140                   | 42.9      | Organic                    |
|     | F            | -0.07 to -0.04 | 220                   | 38.8      | Organic                    |
|     | Н            | -0.04 to 0.00  | 386                   | 33.1      | Organic                    |
|     | Ae, Aeg      | 0.00 to 0.21   | 1316                  | 0.66      | Loam (49,41,10)            |
|     | Bt, Btg, Bmk | 0.21 to 0.69   | 1502                  | 0.32      | Sandy Clay Loam (49,27,24) |
|     | Ck           | 0.69 to 0.83   | 1438                  | 0.31      | Sandy Clay Loam (54,24,22) |
| OBS | Peat**       | -0.20 to -0.10 | 44                    | 42.6      | Organic                    |
|     | Peat**       | -0.10 to 0.00  | 120                   | 44.9      | Organic                    |
|     | Ae           | 0.00 to 0.02   | 1250                  | 1.40      | Sandy loam (76,20,4)       |
|     | AB           | 0.02 to 0.05   | 1520                  | 0.98      | Sandy loam (73,21,6)       |
|     | Bt           | 0.05 to 0.17   | 1660                  | 0.36      | Sandy loam (64,29,7)       |
|     | Bfj          | 0.17 to 0.42   | 1660                  | 0.10      | Sand (96,2,2)              |
|     | Ck           | 0.42 to 0.72   | 1660                  | 0.05      | Sand (96,2,2)              |
| OJP | LFH          | -0.02 to 0.00  | 243                   | 25.1      | Organic                    |
|     | Ae           | 0.00 to 0.02   | 1225                  | 1.00      | Sand (94,3,3)              |
|     | AB           | 0.02 to 0.06   | 1447                  | 0.64      | Sand (93,4,3)              |
|     | Bm           | 0.06 to 0.34   | 1483                  | 0.13      | Sand (94,3,3)              |
|     | C1           | 0.34 to 0.83   | 1517                  | 0.02      | Sand (96,2,2)              |

# Table 2 Soil properties at the BERMS flux towers, by horizon\*.

\* excerpted from Anderson et al. (2000).

\*\* At OBS, the depth of the peat layer within the flux-tower footprint is highly variable, in some places approaching 1.0 m; these estimates are from the locations of the soil temperature and water content profiles and may be shallower than the mean depth.

# **3** Instrumentation and Measurements

A list of the core instrumentation is included in Table A1.

# 3.1 Air temperature, humidity, pressure, and wind

Air temperature  $T_a$  (°C) and relative humidity RH (%) were measured at three or four heights at each site using temperature/humidity probes (models HMP35CF, HMP45C, and HMP155A, Vaisala Inc., Oy, Finland) mounted 30 cm away from southeast corner of the flux tower in 12-plate (or 14-plate for HMP155A) unventilated Gill radiation

shields (models 41002 and 41005, Campbell Scientific, Logan, UT, USA). At the forested sites, the measurement heights included above-canopy (approximately twice the canopy height – 37 m at OA, 25 m at OBS, and 28 m at OJP)

and within-canopy locations (1, 4, and 18 m at OA, 1 and 6 m at OBS, and 1 and 10 m at OJP). At the uppermost level, air temperature was also measured using unshielded fine-wire type-E chromel-constantan thermocouples (25 or 75 µm), and a shielded and ventilated thermocouple and platinum resistance thermometer (mode RTD-810, Omega

Engineering, Norwalk, CT, USA) in an aspirated radiation shield (model 076B, Met One Instruments, Grants Path, 170 OR, USA). The measurements in the aspirated radiation shield had frequent dropouts and periods when the fan failed and so were used only in filling gaps when the other measurements failed.

The RH data were corrected to ensure that the maximum value did not exceed 100%. The measured RH maximum varied among sensors; in particular, some of the earlier model HMP35CF sensors routinely reached a maximum RH

as high as 110%. A simple correction was applied based on a sensor-specific estimate of the upper RH envelope 175 (RH<sub>x</sub>), estimated as the 99<sup>th</sup> percentile of the warm-season measurements (based on air temperatures of between 0 and 20 °C, the range over which RH reached 100%). The corrected value RH\* was computed as:  $RH^* = (RH/RH_x)*100\%$ , (1)

Wind speed and direction were measured at the top of the flux tower using a propeller anemometer (model 05103,

R.M. Young, Traverse City, MI, USA). Atmospheric pressure  $p_a$  (kPa) was measured in the instrumentation hut at ~2 m above ground level using a barometric pressure sensor (model SBP270, Setra Systems Inc., Boxborough, MA, USA at OA, OBS, and OJP, and model PTA427A, Vaisala Inc., Oy, Finland at Fen). Vapor pressure  $e_a$  (kPa) was computed from RH and the saturation vapor pressure over water  $e_{sa}$  (kPa), which in turn

was computed as a function of  $T_a$  using the six-parameter formulation of Hyland and Wexler (1983), given as Eq. 2.5 in Flatau, Walko and Cotton (1992). Specific humidity  $q_a$  (kg kg<sup>-1</sup>) was then calculated from  $e_a$  and  $p_a$  as: 185  $q = 0.622e_a/(p_a - 0.378e_a),$ (2)

# **3.2 Precipitation**

Precipitation was measured at OA, OBS, OJP and Fen using an accumulating gauge (Belfort model 3000 (Belfort Instruments, Baltimore, MD, USA) for all years at OA, 1997 to 2011 at OBS, 1997 to 2010 at OJP, and 1997 to 2010 at Fen; and Geonor model T-200B (GEONOR Inc., Augusta, NJ, USA) at OBS starting Jan 2012, OJP starting Sept

height from the clearing edges, to minimize wind effects on snowfall catch efficiency. At OA, the gauge was mounted

2010, and Fen starting Jan 2011. The resolution of the Belfort 300 gauge was 1 mm, whereas the resolution of the Geonor T-200B was less than 0.01 mm, as determined by the data-logger resolution. At the three forest sites, the gauges were situated in the centre of small forest clearings, at approximately one tree

- on a raised platform at ~ 3 m above ground level (agl), ~50 m northeast of the flux tower. At OBS, the Belfort gauge (and the replacement Geonor gauge) were mounted on the roof of the instrument hut until May 2023, at ~5 m agl. In May 2023, the Geonor gauge was moved off the hut and mounted on a pedestal at  $\sim 2 \text{ m}$  agl. At OJP, the Belfort gauge was mounted on a raised platform at ~3 m agl, ~150 m southeast of the flux tower. The replacement Geonor gauge was mounted on a pedestal at ~2 m agl, ~125 m east of the flux tower. At Fen, up to 2011, the Belfort gauge was
- mounted on a pedestal in the open fen at ~2 m agl, where it was exposed to the wind. The replacement Geonor gauge was installed in Sept 2010 in a sheltered forest clearing atop the instrument hut at ~5 m agl. In May 2023, it was moved to a nearby location in the forest clearing and mounted on pedestal at ~ 2 m agl. In all cases, the precipitation

gauges were equipped with a single Alter shield to reduce wind effects on snowfall catch efficiency. A tipping bucket rain gauge (Texas Instruments model TE525 then Hydrological Services TB3) was also deployed near each

205 accumulating gauge, and was used to fill gaps and to determine rainfall event timing for the Belfort gauge which had only 1 mm resolution. A propeller anemometer (model 05103, R.M. Young Co, Traverse City, MI, USA) or 3-cup anemometer (model 12102, Gill Instruments Ltd., Lymington, UK) was installed nearby to measure windspeed at gauge height.

The accumulating gauges were serviced twice a year. Each spring, the bucket was charged with 1.5-2 L of water and 500 mL of 0W-20 motor oil. Each fall, the bucket was charged with 4 L of antifreeze and 500 mL of 0W-20 motor oil. Motor oil was added year-round to minimize evaporative losses. Antifreeze was added to prevent freezing of the

bucket contents in the winter months. The cumulative precipitation time series was smoothed using a neutral aggregating filter (Ross et al., 2020) to

eliminate noise and extract the 30-min interval precipitation amounts, aggregated to a minimum value of 0.1 mm. The

- algorithm removes random noise and accounts for diurnal oscillations in the bucket weight signal but does not account for drift, which means that it will not perform well if the accumulating gauge has periods of water loss by evaporation (Smith et al., 2018). Precipitation was then partitioned into rainfall and snowfall based on air temperature and humidity (Harder and Pomeroy, 2013). At the three forest sites and at Fen after 2010, the gauges were sheltered from the wind and no adjustments were needed to correct for wind-induced undercatch of snowfall. At the Fen from 2003
- to 2010, when the Belfort gauge was situated in the open and exposed to wind, measured snowfall was corrected for wind-induced undercatch based on the catch efficiency CE, estimated from the measured windspeed U (m s<sup>-1</sup>) and air temperature  $T_a$  (°C) at gauge height using the empirical transfer functions developed by the WMO Solid Precipitation Intercomparison Experiment (SPICE) (Smith et al., 2020, equation 1):

$$CE = \begin{cases} e^{-aU(1-\tan^{-1}(bT_a)+c)}; \ T_a \le T_a^* \\ 1; \ T_a > T_a^* \end{cases}$$
(3)

where: a = 0.0348; b = 1.366; c = 0.779; and  $T_a^*$  was set to 2 °C. The CE adjustment at the Fen increased annual snowfall by an average of 34% and annual *P* by an average of 7%, 2003-2010. Snow depth (continuous) and snow water equivalent (periodic snow surveys) were also measured but are not included in this data set.

#### 3.3 Soil temperature and water content

- Soil temperature  $T_s$  was measured using copper-constantan (type T) thermocouples at two locations, with vertical profiles of six depths (2, 5, 10, 20, 50, 100 cm) below the ground surface (relative to the top of the upper organic soil horizon, Table 2). Vertical soil volumetric water content VWC profiles were measured at two or more locations per site, but each site had its own configuration of sensors and depths. OA used time-domain-reflectometer TDR probes at depths of 0-15, 15-30, 30-60, 60-90, and 90-120 cm. Up to 2007, the measurements were made using four profiles
- of Moisture Point MP-917 type-B probes (Environmental Services Inc., Victoria, B.C., Canada) located in a low-lying area of the site. That location partially flooded in 2004 and remained flooded through 2007, and so in late summer 2007 three profiles of CSI TDR100 probes (Campbell Scientific, Logan, UT, USA) were installed at a new location