# Peer review of "Long-term meteorological and carbon, water and energy flux data from the Boreal Ecosystem Research and Monitoring Sites, Saskatchewan, Canada"

_Earth System Science Data, 2024_

## Author Comment (AC1)

**Response to reviewers for essd-2025-492 Long-term meteorological and carbon, water and energy flux data from the Boreal Ecosystem Research and Monitoring Sites, Saskatchewan, Canada by Barr and coauthors.**

Both reviewers put a lot of effort into their reviews, and we thank them for recommendations that are thoughtful, and constructive. The two reviewers make several of shared recommendations that we have incorporated into the revision. Two areas in which they diverge are in QA/QC and the resolution of known problems. Reviewer 1 recommends including further detail whereas Reviewer 2 sees these as an area of strength. We attempt to reconcile these differences as we provide individual response to each point on the following pages.

Reviewer comments are identified by black text in italics.

Author comments can be identified by red text.

**Response to Reviewer #1**

*RC1: The manuscript provides a substantial documentation of long-term dataset from the Boreal Ecosystem Research and Monitoring Sites in Saskatchewan, Canada. It includes site-level flux measurements of carbon, water, and energy from four vegetation types over nearly three decades. The data are intended to support ecological modeling, inter-site comparisons, and Earth system analyses. While this manuscript provides a significant dataset for future research, it can benefit from substantial improvements as follows:*

- *The overall purpose of the paper is not introduced until deep into the text. The abstract and early sections lack an explicit statement of aim—specifically, to describe dataset content, quality, and usability. This should be clearly expressed in the first or second paragraph.*

  AC1: The introduction section has been condensed and considerably shortened, thus allowing the aim of the paper to appear more prominently in this section. Please see the tracked changes version.

- *Details regarding quality control, gap-filling methods, and flagging protocols are currently too summary. For reproducibility and transparency, include specifics on data cleaning routines, error thresholds, and imputation strategies.*

  AC1: Our judgment is that the QA/QC and gap-filling methods are adequately documented in Sections 3.6, Appendix B, and Sections 3.7.6 and 3.7.8. We acknowledge that the documentation in this paper – and the data available to the reader -- will not enable the reader to start from scratch and reproduce our processing exactly so that every datum is identical. One simple reason is that they do not have access to the raw data but only to the final, processed data product. Another reason is that our QA/QC approach includes some manual intervention as described in Section 3.6 line 275:

  > "All meteorological data were screened daily using automated limit checking followed monthly by manual, graphical inspection and identification and exclusion of questionable values. The exclusion criteria were conservative, so that only extreme outliers were flagged and excluded."

  If the reader wishes to repeat the manual outlier exclusion, a few differences may result from different subjective choices in outlier identification. But the differences will be subtle; the reprocessed dataset will be essentially consistent with our processing.

- *The manuscript would be strengthened by providing a concise data schema or table summarizing variables, units, temporal resolution, and quality flags. More information on file formats, naming conventions, and metadata components is essential.*

  AC1: Thank you for this suggestion. To provide more information to the reader regarding the specific dataset we have included a new section *6.1 Data specification* which describes the temporal frequency and the duration of the records. We have also created a new *Appendix F* which contains a table specifying the specific variables included in the dataset for each site. The archived dataset includes a metadata file that provides the variable naming conventions for each file so those are not repeated in this paper.

- *Known problems or limitations with the data are acknowledged, but resolution strategies or remaining caveats are not sufficiently detailed. Present any unresolved issues, their implications, and planned updates or revision mechanisms.*

  AC1: Thank you for noting our efforts to highlight problems that we have identified in the data. We have resolved all of the major issues that we identified to the extent that our resources will allow; no further actions to modify the related data are proposed. In the case of the discontinuity potentially caused by the change in flux measurement instrumentation, we have carefully documented (Section 5.1 and Appendix D) the stability of the energy closure fraction, the friction velocity threshold, and the random uncertainty of NEE. Here we make the reader aware of the issue and any consequences, but there are no data revision procedures which can be implemented. In the case of noticing that the PPFD sensors have deteriorated with age, we describe a site-specific calibration that we use to fill these data in Appendix E). Similarly, with VWC sensors that have stopped working over the years, we describe the issue in section 5.2 and present more details in Appendix E2 as to how the resulting VWC time series in the published dataset have been constructed.

- *The manuscript would benefit from a clearer discussion of how the dataset enhances or complements current global flux networks (e.g., AmeriFlux, FLUXNET). Highlight how these data fill geographic or temporal gaps.*

  AC1: We agree. We have now added section *6.2 Relationship to other datasets* which describes the other previous, or related, datasets, including those in the Ameriflux database, and lists how the data presented in this manuscript are different.

*The dataset is highly valuable and aligns well with the mission of Earth System Science Data. However, to reach publishable quality, the manuscript requires clearer articulation of aims, stronger integration with existing flux networks, and more comprehensive data documentation, quality control, and usability guidance.*

Major points

AC1: Responses to these 5 Major points have been addressed through the preceding comments.

1. *Re-structure the introduction to state paper's objectives upfront.*

2. *Expand on QA/QC and data processing protocols.*

3. *Provide concise data tables and metadata schema.*

4. *Integrate comparisons with other flux networks.*

5. *Clarify known limitations and plans.*

*Minor points*

- *The abstract effectively lists what the dataset contains but lacks a clear statement of purpose. Add a sentence explaining why the dataset is being published and how it supports Earth system science research. Ensure subject-verb agreement (e.g., 'dataset include' should be 'dataset includes') and streamline awkward phrasing.*

  AC1: The following sentence has been added to the abstract: "This multi-year dataset enables inter-site comparisons of carbon and water balance responses to climatic controls, it provides driving and validation data for process-based ecosystem and hydrological models and provides opportunities for ground-truthing remotely-sensed products of ecosystem function." The revised abstract has also been reviewed and edited for any grammatical and phrasing issues.

- *The introduction sounds more like a description of the study site rather than the dataset and why is it important. Discuss the scientific significance of long-term boreal flux data and its role in addressing current climate and ecosystem science challenges.*

  *AC1: We have moved some of the description of the boreal plains ecozone to the study site section. The introduction section was substantially revised so that the objectives of the observation network, the aim of this data description paper, and the potential significance and uses of the data are more prominent.*

- *RC1: Figures abbreviations should be expanded for all figures. Consider moving the non relevant ones to a supplementary figure.*

  AC1: Figure abbreviations have now been expanded for all figures.

- RC1: Remove subjective descriptors such as "the care given to..."; instead, use neutral and objective phrasing (e.g., "instrumentation provided high-precision measurements").

- AC1: This has been done.

- *RC1: Review for minor grammar errors—e.g., change "dataset include" to "dataset includes"—and streamline dense sentences for readability.*

  AC1: The revised version has been carefully reviewed for grammatical mistakes and corrected (see tracked changes).

**Author Response to Reviewer # 2**

*RC2: General comments*

*This manuscript presents a long-term dataset of meteorological and flux measurements from four sites of the Boreal Ecosystem Research and Monitoring Sites in Saskatchewan, Canada. Long-term datasets spanning multiple sites and ecosystem types are highly valuable for investigating ecosystem responses to environmental drivers, including climate change, and are of great importance for model evaluation and calibration. The description of the study sites, instrumentation, and measurements is thorough, and the discussion of measurement uncertainties and potential issues is well developed.*

*However, the data overview section is overly descriptive, especially for the energy, carbon, and water fluxes. The manuscript would benefit from a stronger integration of these results with the site-specific characteristics, as outlined in my "Specific comments" below. In addition, I find that the significance of the dataset is not sufficiently highlighted: (i) the introduction should more explicitly state the objectives of the paper, and (ii) the potential applications of the dataset could be further developed, either in the conclusion or in a dedicated section preceding it.*

AC2:

- The introduction section has been shortened with some ancillary description of the boreal plains ecozone moved to the study site section. The key objectives of the paper have been made more clear as well as the potential uses for the data. This has been strengthened in the introduction as well as in the conclusions.
- We have not implemented all of your suggestions to integrate additional site-specific characteristics into the manuscript, as the scope of the journal limits the amount of data interpretation that we can provide here. We wholeheartedly agree that site-specific analysis would be worthy of further investigation and we hope that by providing these data it will enable researchers from the community to explore these sites through more thoroughly.

*Overall, this manuscript fits well within the scope of the journal, and I recommend publication after addressing the points below.*

*RC2 Specific comments*

*Figure 1: Add the latitude and longitude ticks. Could you represent the Boreal Plains Ecozone area on the right map?*

AC2: A new figure 1 has been included which has latitude and longitude marks and identifies the boreal plains ecozone in relation to the prairie and boreal shield ecozones.

L93: "27% of which fell as snow (based on the 1981-2010 normals)": Could you clarify why the 1981-2010 normals are used here instead of the same period as for precipitation (1991-2020)?

AC2: This has now been corrected so that the 1991-2020 normals were also used for snow. Interestingly, the portion of snow changed from 27% to 20% during this period.

L232: "Vertical soil volumetric water content VWC profiles were measured at two or more locations per site, but each site had its own configuration of sensors and depths." Could you clarify why the same soil depths were not used across sites for VWC measurements?

AC2: We have tried to clarify this by modifying that sentence to also include the rationale for the site-specific configuration. The proposed sentence is: "Vertical soil volumetric water content (VWC) profiles were measured at two or more locations per site, but each site had its own configuration of sensors and depths; reflecting the unique soil structure and moisture conditions."

L424: "An analysis of the stability of the u*TH over time is provided in Appendix C." Since Appendix C shows that annual u*TH values have an impact on flux estimates and exhibit temporal trends, it may be worth noting here that using fixed u*TH thresholds could limit the interpretation of interannual variability or long-term trends in carbon fluxes.

AC2: This is a good point. We hope that the data users will review the material in Appendix C to better understand the nature of this correction. Nonetheless, we have added the following sentence to section 3.7.6 Quality Assurance: "It is acknowledged that using a constant value of $u_*^{TH}$ over the period of record may limit the interpretation of interannual variability or long-term trends of carbon fluxes."

L460: Why was CF derived annually rather than at shorter timescales? Did you check whether CF shows seasonal variation at each site? This could be relevant, since model evaluation for example usually requires energy balance closure throughout the year rather than only as an annual adjustment.

AC2: The CF value was derived annually based on the warm season data only. CF cannot be reliably measured during the winter periods because we do not measure stored internal energy within the snowpack. Support for the method we used here can be found in Barr et al. (2006) and Barr et al. (2012). We acknowledge that this may not meet the needs of all users, so within the dataset we provide evapotranspiration with energy balance closure and also without closure.

Barr, A. G., Morgenstern, K., Black, T. A., McCaughey, J. H., & Nesic, Z. (2006). Surface energy balance closure by the eddy-covariance method above three boreal forest stands and implications for the measurement of the CO2 flux. *Agricultural and Forest Meteorology*, *140*(1–4), 322–337. https://doi.org/10.1016/j.agrformet.2006.08.007

L499: *"likely related to changes in surface energy partitioning with increasing soil water deficits in July and Aug." Could you provide a figure in the appendix to support this statement?*

AC2: This statement has been removed. Our intention with this paper was just to provide a brief overview of the data rather than interpretation.

L501-505: *This section could be more analytical rather than purely descriptive. For example, the lower correlation of VPD anomalies with wind speed anomalies, compared to air temperature and shortwave radiation anomalies, could be discussed in terms of the primary drivers of VPD.*

AC2: We consider this degree of analysis (interpretation) to be beyond the scope of the submission.

L551-554: *Could you mention the implications of the Q/Rn ratios in terms of site-specific characteristics, such as vegetation structure or soil properties? Also, expressing these ratios as percentages could improve readability.*

AC2: We have changed these ratios to percentages. However, we feel that attributing the percentages to site-specific characteristics would require interpretation and be beyond the scope of this submission.

L561-568: *Similar to previous comments, this section is mostly descriptive. It would be helpful to discuss the observed differences in H and λE among sites in terms of vegetation functioning, canopy structure, or site-specific hydrological conditions.*

AC2: This would be interesting to investigate, but we feel the interpretive analysis is beyond the scope.

*Table 4 could be moved to the appendix.*

AC2: Agreed. This has been done (Inserted as new Appendix C)

L645: *Have you looked at the correlation between FC and MDS NEP at daily or weekly timescales? This could provide additional insight into how the two gap-filling methods compare for shorter-term variability.*

AC2: We have not compared FC and MDS NEP at shorter timescales. This is interesting, but beyond the scope of this contribution.

L730-735: Why not provide a quality flag for these questionable winter periods (snow, hoar frost, freeze-thaw), so that users can easily identify and handle them?

AC2: We have not done this and feel that it would be a significant undertaking to do so at this point. Ultimately, we leave it up to users to implement their own flagging system.

*Figure C1: At OJP, the drop in CF coincides with the instrument change from UBC to LCR, but at OBS the drop occurs earlier (around 2004), before the instrument change. Could you comment on what might explain this earlier decline? In light of this, are the comparison periods (1997–2010 vs 2011–2022) still appropriate for OBS?*

AC2:  Unfortunately, we don't know what might explain the earlier decline in CF at OBS. With respect to comparison periods, those are only valid for OJP as the instrument at OBS was not changed until 2018.

**Technical corrections**

*L37: Provide a reference or a link for these data: "The BOREAS data from 1994-1996 are available at the Oak Ridge National Laboratory Distributed Active Archive Center (ORNL DAAC)"*

AC2: This statement has been moved to section 6 and references have been added.

*L41: Provide references or links for these networks: "BERMS played a leading role in the Fluxnet-Canada Research Network (FCRN, 2002-2007) and the Canadian Carbon Program (2007-2011), and has contributed to the Changing Cold Regions Network (2012-2017), Global Water Futures (2017-2023), the Global Water Futures Observatories (starting 2023), the North American Carbon Program, AmeriFlux, and FLUXNET"*

AC2: Done

*L34: Remove "of" in "The primary BOREAS objectives were to better understand **of** the role…"* AC2: Done

*L36: "ecosystem C-cycle models". Define "C" as carbon before using the abbreviation.* AC2: Done

*Table 1: Replace "in error" with "an error": "In particular, the earlier value of 579 m at OJP was found to be **an** error".* AC2: Done

*Table 2: Please specify that the soil depth values are given relative to the mineral soil surface, to clarify the meaning of negative and positive values.*  AC2: Done

*L222: Define "WMO".* AC2: Done

*L233: Please precise: "OA used time-domain-reflectometer TDR probes installed to measure VWC **integrated over the soil layers** 0-15, 15-30, 30-60, 60-90, and 90-120 cm"* AC2: Done

*L299: Replace (1) by (4) in "All terms in **(4)**…"* AC2: Done

*L305: Replace (2) by (5) in "Typical units for the variables in **(5)**…"* AC2: Done

*L392: Add "on" in "based **on** the measured air temperature".* AC2: Done

*L421: Remove the parenthesis at the end of this sentence: "One value was applied across all years: 0.35 m s-1 at OA, 0.30 m s-1 at OBS, 0.25 m s-1 at OJP, and 0.12 m s-1 at Fen**)**."* AC2: Done

*Figure 4 caption: Change the description of the upper panel "green - positive anomalies; red – negative anomalies" with the correct colors.* AC2: Done

*L497: Correct "earlier **in** autumn".* AC2: Done

*L538: Specify "Prince Albert Airport" instead of "Prince Albert A".* AC2: Done

*L646: Define "r".* AC2: Done